# A Novel Method for Stimulating *Cannabis sativa* L. Male Flowers from Female Plants

**DOI:** 10.3390/plants12193371

**Published:** 2023-09-25

**Authors:** Luke C. Owen, David H. Suchoff, Hsuan Chen

**Affiliations:** 1North Carolina Cooperative Extension, Buncombe County Center, Asheville, NC 28803, USA; lcowen@ncsu.edu; 2Department of Crop and Soil Sciences, North Carolina State University, Raleigh, NC 27695, USA; dhsuchof@ncsu.edu; 3Department of Horticultural Sciences, North Carolina State University, Raleigh, NC 27695, USA

**Keywords:** STS, feminized seed, hemp, hemp breeding, ethylene inhibitors, pollen viability

## Abstract

Female hemp plants are desired in floral hemp operations due to their higher cannabinoid contents. To produce feminized seeds, a critical step of inducing fertile male flowers on female plants is performed. In feminized seed production, freshly mixed STS (silver thiosulfate + sodium thiosulfate) is applied to female plants as an ethylene inhibitor to induce male flowers. However, the short-shelf stability of the STS buffer can cause difficulty in the application and inconsistent results. Alternative methods with improved accessibility and stable buffers will be beneficial for the hemp industry and hemp breeders. A commercially available floriculture product, Chrysal ALESCO^®^, contains silver nitrate, the same active ingredient as STS but with increased shelf stability. This study compares Chrysal ALESCO^®^ to the traditional STS standard methods for male flower induction on female plants and their pollen quality. The two treatments were applied to six female hemp accessions with three replicates investigated, and the male flower counts and pollen quality were compared. No statistically significant difference was discovered in their male flower counts; the STS-treated plant produced an average of 478.18 male flowers, and the Chrysal ALESCO^®^-treated plant produced an average of 498.24 male flowers per plant. Fluorescein diacetate (FDA) and acetocarmine stains were used to investigate the pollen quality (non-aborted rate) of two chosen genotypes. FDA-stained pollen of Chrysal ALESCO^®^ showed a significantly higher non-aborted rate than the pollen of traditional STS-treated plants (*p* < 0.001); however, only a marginally higher non-aborted rate was discovered by acetocarmine staining (*p* = 0.0892). In summary, Chrysal ALESCO^®^ performed equally to traditional STS treatment at male flower counts and better or equally in pollen quality. With better shelf stability and easy application, ALESCO^®^ can be a viable alternative option for stimulating male flowers on female hemp plants.

## 1. Introduction

Hemp (*Cannabis sativa* L. < 0.3% *w*/*w* total tetrahydrocannabinol [THC]) has been an increasingly important economic crop in the United States since the 2018 Farm Bill [1,2]. Hemp is naturally a diploid (2n = 2x = 20) dioecious species whose plant sex is determined by sex chromosomes, with heteromorphic (XY) signaling a male plant and homogametic (XX) signaling a female plant [3]. When grown for cannabinoids, only female plants are selected due to the synthesis of high concentrations of cannabinoids stored in glandular trichomes [4]. Male plants produce comparatively low amounts of cannabinoids [3,5]. Furthermore, pollination can drastically decrease the cannabinoid contents of female flowers by as much as 56% [6]. Initially, producers sourced high-cannabinoid female hemp cultivars from asexually propagated clones, and, in 2021, the total value of the production of hemp clones was $23.8 million, with a total of 20.2 million clones produced in the United States [7]. On average, the cost of asexually produced female floral hemp cultivars ranges from $5.00–10.00 per clonal propagule [8]. Female plants pollinated by genetically female pollen allow for the production of female seeds (“feminized seeds”) [9]. Much of the industry has shifted to developing or using feminized seed, allowing growers to reduce input costs, reduce plant disease risk, and increase growth vigor from the plant juvenile phase. The average cost of feminized seed from select hybrid lines is $0.50–1.00 per seed, depending on the amount ordered, source, and cultivar selected [8]. Therefore, producing feminized seeds is crucial to the floral hemp industry since only females are used in commercial production [9]. To produce feminized seeds, artificially stimulating male flowers and pollen production on genetically female plants becomes a critical step. Pollen that is genetically female does not carry the Y chromosome and can only result in female seeds.

Exogenous application of plant growth regulators (PGRs) and their inhibitors have been used to regulate and alter flower sex expression in many crops, including *Cucumis sativus* [10], *Silene latifolia* [11], and *Coccinia grandis* [12]. Specific PGR applications include gibberellins, which have been shown to favor male flower expression, while ethylene, cytokinin, and auxins all stimulate the formation of female flowers [9]. In hemp, exogenous application of plant growth regulators has been commonly used to produce male flowers from female plants for breeding purposes or to produce feminized seeds [9,13,14,15,16,17,18,19]. Two silver-based ethylene inhibitors, silver thiosulfate (STS) and colloidal silver and gibberellic acid (GA) spray, have been commonly applied for triggering male flowers on female *Cannabis* plants [9]. The STS method has shown the best efficiency in stimulating male flower counts [9] and has been commonly used in commercial feminized seed production and breeding processes [20].

Ethylene inhibitors are used in the horticultural industry to regulate flower senescence, increase fruit and flower shelf life, and alter plant habits [17]. Ethylene inhibitors operate through two primary mechanisms: inhibition of ethylene precursor biosynthesis or disruption of ethylene action via ethylene receptor binding [17]. The STS is one of the most commonly used ethylene inhibitors. The STS foliar spay is the product of silver nitrate and sodium thiosulfate. It has been used to initiate male flowers by inhibiting ethylene signals by breaking down the hormone into inactive compounds [21]. Ethylene receptors in the plant require copper co-factors for correct binding. When this binding fails, it allows physical interaction with CONSTITUTIVE TRIPLE RESPONSE (CTR1), a Raf kinase, to repress the positive feedback regulator ETHYLENE INSENSITIVE2 (EIN2), which produces ethylene [15,16,19]. Silver thiosulfate inhibits ethylene production by outcompeting copper ion cofactors in the ethylene receptors, allowing CTR1 to be active and signaling a slowed ethylene response, which initiates male flower production [19]. Through this mechanism, STS inhibits ethylene signals and stimulates male flower initiation.

The effectiveness of PGR application in hemp can be measured by the total number of male flowers and the quality of the resultant pollen. Male flowers are a direct comparative index of the PGR application effectiveness; however, a universally accepted standard method to estimate hemp quality is not yet available. Pollen germination and pollen staining methods are frequently used to investigate pollen qualities, and each method has its advantages and drawbacks. Pollen in vitro germination provides strong evidence of pollen viability. However, optimal methods might only be available for some plant species, but not yet for hemp. In hemp, the most updated pollen in vitro germination method showed only 30–50% germination of fresh pollen, and the rate significantly depended on the mother plant genotypes [21]. Another study compares the pollen viability of different genotypes through in vitro pollen germination rates ranging from 0 to 19% [22]. On the other hand, pollen staining methods that test the pollen non-abortion rate can be used to compare the qualities of pollen. Still, it might only partially indicate the real pollen viability (or fertility). Fluorescein diacetate (FDA) and acetocarmine staining are the most commonly used methods to investigate pollen quality (non-abortion rate). Fluorescein diacetate stains intact semi-permeable membranes, and intracellular esterases can hydrolyze FDA in the cytoplasm of the pollen cell, showing bright green inflorescence [23]. If the plasmalemma is not intact, the fluorescein does not accumulate, indicating lower-quality malformed pollen granules [24,25,26]. Acetocarmine stains chromatin located within the pollen nuclei, indicating pollen that is fully developed or has a complete structure [26]. In hemp, all three methods have been used in pollen quality investigations [21,22,27,28]; some are specifically used to compare pollen quality from different male flower stimulation methods [9].

There are several drawbacks to the traditional STS method when used in commercial agriculture environments. Traditional STS lacks shelf stability, requiring a freshly prepared stock solution to induce male flowers for every spray event [15]. The lack of shelf stability could increase the labor required because the buffer must be freshly made for each of the three applications [18,20]. The unstable silver nitrate in the STS can precipitate before foliar spraying, resulting in potentially inconsistent male flower stimulation. Chrysal ALESCO^®^ Anti-Ethylene Treatment (ALESCO^®^) is a commercially available product for use as a potted plant conditioner. This product was originally used to avoid stress-induced premature shrinking or dropping of buds or blooms on ethylene-sensitive ornamental crops and to extend the shelf-life of ornamental products [29]. Since both traditional STS Spay and ALESCO^®^ target the blocking of ethylene-sensitive molecules by the same active ingredient, silver thiosulfate, using the shelf-stable commercial product, ALESCO^®^, to replace traditional STS could be a worthwhile strategy.

This study is to evaluate the potential and efficiency of using ALESCO^®^ to replace the traditional STS foliar spray to stimulate male flower stimulation from female plants. In the current study, *C. sativa* plants were treated with the two traditional STS and ALESCO^®^ separately, and the effectiveness of male flower initiation (flower count per plant) and the quality of pollen were investigated and compared.

## 2. Results

### 2.1. Ethylene Inhibitor’s Impact on Flower Count

The average male flower production across all treatments was 473 male flowers per plant. Plants from the traditional STS treatment produced an average of 478.18 male flowers per plant, and plants from the ALESCO^®^ treatment produced an average of 498.24 male flowers per plant (Figure 1). Two plants in the exam were dead before they flowered: one STS-treated H2020-376-002 plant and one ALESCO^®^-treated H2020-376-002 plant. No significant ethylene treatment, genotype, or interaction effects on male flower count (*p* > 0.05) existed. The flower counts of each plant were listed in Appendix A.

### 2.2. Pollen Quality (Abortion Rate) by FDA and Acetocarmine

Fresh pollen of two different accessions, H2020-425-015 and H2020-376-003, were collected, and both FDA and acetocarmine stain were tested to determine pollen quality. Under each ethylene inhibitor, genotype, and stain method, 697–1397 pollen grains were investigated (Figure 2). A total of 9142 fresh pollen grains were investigated, and an average of 86.32% were non-aborted. By the FDA stain, 4355 pollen grains were investigated, and 3718 (85.37%) grains were non-aborted. For the 2% acetocarmine stain, 4787 pollen grains were investigated, and 4173 (87.17%) grains were non-aborted.

Using the FDA stain, H2020-376-003 treated by ALESCO^®^ and STS showed 94.44% and 88.33% non-abortion rates, and H2020_425_015 treated by ALESCO^®^ and STS showed 78.17% and 71.17% non-abortion rates. ALESCO^®^ showed a statistically significant increase in pollen non-abortion rate (*p* < 0.001) in the logistic GLM analysis. By the 2% acetocarmine stain, H2020-376-003 treated by ALESCO^®^ and STS showed 90.86% and 92.29% non-abortion rates, and H2020_425_015 treated by ALESCO^®^ and STS showed 84.76% and 79.78% non-abortion rates. ALESCO^®^ showed a moderately significant increase in pollen non-abortion rate (*p* = 0.0892). Pollen quality (non-abortion rate) from ALESCO^®^-treated plants was generally higher than STS treatment. The result is listed in Figure 3 and Appendix A.

## 3. Discussion

### 3.1. Chrysal ALESCO^®^ in Male Flower Count and Pollen Quality

Research and industry have been working on discovering the optimal hemp male flower stimulation methods for breeding processes and industrial-scale feminized seed production. Repeatedly, the STS treatment has been scientifically reported as a very effective method [9,18]. Other methods, like gibberellic acid or colloidal silver sprays, have become less used because of their low efficiency or difficulties in their applications. Gibberellic acid spray on hemp showed low efficiency in male flower stimulation; only 2.2% of inflorescent showed male flowers, and STS spray resulted in 21.0–53.8% of inflorescent showing male flowers [9]. The colloidal silver spray must be applied daily to produce only 55–60% of the male flower counts of STS-treated plants [9]. The other sliver compound, AgNO_3_, was also compared to STS treatment comprehensively, and AgNO_3_ produced a significantly reduced male flower count [14]. Aminoethoxyvinylglycine (AVG) could also be used to stimulate male flowers from female hemp plants; however, there is a lack of research on their effectiveness compared to other methods [30]. Our results, comparing traditional STS and ALESCO^®^ spray, showed that ALESCO^®^ spray has equal effectiveness in stimulating male flowers (flower counts per plant; Figure 1) with improved or equal pollen quality (non-abortion rates; Figure 3).

The drawback of STS has not been fully discussed. In a production facility, the instability of STS could be a big issue. Generally, for applying STS, the solution must be made fresh for each application before each of the three treatments to prevent precipitates. Silver thiosulfate is a light-sensitive and easily precipitating chemical, which could then be discovered on devices used for spraying or storage containers. Considering that the silver compound is the active ingredient for male flower stimulation, the instability and precipitation of silver ions in the STS buffer can lead to inconsistent results. Thus, recommending a shelf-stable commercial product for the hemp seed industry and breeders is valuable.

Hassan et al. recommended using 1-methylcyclopropene to replace STS to improve the shelf life of miniature potted rose cv. Amore because of the potentially heavy metal toxicity [31]. For the potted plant, STS was directly sprayed on the commercial products, so the heavy metal toxicity could go to customers’ households with the plants and pots. However, for hemp seed production, STS was only sprayed on the pollen parent plants, so the chance or dosage of active silver transferred from pollen to seeds could be very low. In addition, the feminized seeds from those STS-stimulated pollens are mostly used for growing flowering plants instead of being directly used for culinary purposes. Thus, metal toxicity does not directly impact the production safety of hemp. Considering that 1-methylcyclopropene and silver thiosulfate are all compounds that block the ethylene pathways, 1-methylcyclopropene could also have the potential to stimulate hemp male flowers. However, there is no research on using 1-methylcyclopropene in hemp male flower stimulation. Future research on other types of anti-ethylene chemicals in hemp male flower induction will be needed.

### 3.2. Pollen Quality (Pollen Non-Abortion) Exam

FDA and acetocarmine stains were used to investigate the qualities of pollen from the two ethylene inhibitor treatments. Generally, with one exception, pollen from ALESCO^®^ showed an increased non-abortion rate compared to pollen from the traditional STS treatment (Figure 3). FDA-stained pollen from ALESCO^®^ treatment showed consistently about 6% increased non-abortion rate among the two genotypes, H2020-376-003 and H2020-425-015. However, by acetocarmine stain, ALESCO^®^ treatment consistently showed a 5% increased non-abortion rate in H2020-425-015 but a 1% reduction in H2020_376_003, compared to the traditional STS treatment (Figure 3. Generally, pollen from the ALESCO^®^ treatment showed better quality.

Pollen stains can be used to compare pollen quality, but not necessarily viability or the percentage of pollen that can pollinate female flowers and produce seed. There is no “golden index” to represent hemp pollen’s viability in producing seeds. Methods used in investigating hemp pollen quality include the number of developing seeds after test pollination [9], in vivo pollen germination [9], in vitro pollen germination [9,21,22,27], and different pollen stains [9,21,27]. Advantages and drawbacks are shown for each method. Seed number after test pollination could be the strongest evidence of pollen viability; however, standardizing pollen number and evenly spraying pollen on female flowers is difficult. Thus, this method directly proves pollen viability, but it would not be feasible to accurately estimate the level of viability. On the other hand, in vitro and in vivo pollen germination tests could be direct evidence of pollen viability. However, no optimal protocol for hemp pollen is available; low germination rates were observed in all research [9,21,22,27], and the germination rates relied on the genotypes of the plant [21,22]. It is hard to say if the germination rate reflects the true pollen viability in producing seeds or just the fitness of the genotype to the pollen germination method or medium. The low detectable pollen germination rates could lead to underestimating the real pollen viability to produce seeds. For example, in the Flajšman research, pollen produced from plants with a 30 ppm colloidal silver daily treatment showed 0 pollen germination rate, but it produced the highest number of seeds compared to pollen with higher germination rates observed from other treatments [9].

Pollen staining methods could estimate the rate of non-aborted pollen for hemp with the risk of overestimating pollen viability. Wizenberg’s research indicated that the pollen stain (non-abortion rate) result is not significantly correlated to the pollen germination rate (pollen viability) [27]. However, the correlation between the pollen staining rate and the number of seeds produced remains unknown. Considering pollen stain methods directly reflect the rate of well-developed pollen grains by detecting the existence of chromatin [26] or alive cytoplasm [23,27], we recommend that the pollen stain methods be used in comparing the relative quality of pollens from different sources, but not for the true viability of producing seeds.

### 3.3. Other Effects Might Improve Male Flower Stimulation

Our result showed that the traditional STS and ALESCO^®^ spray resulted in the same male flower count per plant and equal or improved pollen quality. Considering the active ingredient of both methods is silver nitrate with the same concentration, one possible exploitation of the different results is the difference in surfactant, wetting agents, or adjuvants. Any one of those compounds could potentially help the active ingredient better touch or enter plant tissue. There is no such agent added to the traditional STS spay. However, the commercial ALESCO^®^’s inactive ingredients are not fully labeled. The difference between the results of traditional STS and ALESCO^®^ could result from any surfactants, wetting agents, or adjuvants. Theoretically, any additive agent that enhances the sliver’s ability to attach and penetrate plant tissues can increase the efficiency of male flower stimulations. We recommend that research in the testing of different surfactants, wetting agents, adjuvants, or mixtures is needed. With better penetration of the active ingredient into the plant tissue, reduced concentration or better male stimulation effectiveness would be expected.

## 4. Materials and Methods

### 4.1. Plant Materials

Six cannabidiol (CBD) hemp accessions sourced from the Mountain Crop Improvement (MCI) Lab located at the Mountain Horticultural Crops Research and Extension Center (MHCREC) in Mills River, NC, were used in this study (Table 1). Clones were selected from a mix of open-pollenated bulk populations generated by several genotypes, including BaOx, Cherry Mon, T1 Iso, and T1. The pedigree is not fully recorded, and the clones were selected primarily by observation of their vigorousness and environmental fitness in North Carolina.

The two ethylene inhibitors investigated were the conventional STS method [20] and the commercially available ALESCO^®^ Anti-Ethylene Treatment (ALESCO^®^; Chrysal International, Naarden, Netherlands) [29]. Three replicates of each hemp accession were used to study the effects of ethylene inhibitor products. The experimental design was a full factorial (two ethylene inhibitor treatments × six hemp accessions) completely randomized with all replicates on the same bench.

This study was conducted in a polytunnel greenhouse maintained at 21 °C at the Horticultural Science Field Lab, Raleigh, NC (35°47′28.9″ N 78°41′53.6″ W) from 10 October 2021 to 7 January 2022. Three replicates of each of the six accessions (Table 1) were propagated from stock plants by taking cuttings for clonal propagation. Stock plants were maintained under 18 h of light (25 µmol m^−2^ s^−1^ BR30 Grow Light Bulbs; Briigite St. Louis, MO, USA) to maintain vegetative growth. Granular slow-release fertilizer (Multicote Extra 8, 18N–3.9P–10K 5- to 6-month formulation; Everris NA, Dublin, OH, USA) was applied at a rate of 2.5 g per container, and a water-soluble fertilizer solution (Peters 20N–8.7P–16.6K; Scotts, Marysville, OH, USA) was hand applied to maintain stock plants as well as cuttings after transplant following product recommendations of 5.6 g L^−1^ solution (Scotts, Marysville, OH, USA). Additionally, a water-soluble calcium and magnesium supplement was added to the solution at a rate of 1.32 mL L^−1^ (Cal-Mag Plus 2.0N-3.2Ca-1.2Mg; Port Washington, NY, USA). The cuttings made had at least three nodes each and were potted into 0.5 L square pots on 29 October 2021, and then moved to four L-sized black plastic containers on 29 November 2021. All treatment replicates were placed randomly on benches under no artificial light on 3 December 2021 to induce flowering during short-day intervals where the natural day length was approximately 9.5 h.

### 4.2. Ethylene Inhibitor Application and Male Flower Count

Each ethylene inhibitor treatment was prepared for a final 1.48 μM silver concentration using a modified protocol from [20]. The total solution volume was 4 L of STS prepared by mixing 0.5 g of silver nitrate and 2.5 g of silver thiosulfate into 1 L of water and thoroughly mixing by adding 3 L of water to make the final 4 L solution. Mixing is required, and the solution must be made freshly for each application to prevent precipitates from forming before treatment. A total of 4 L of ALESCO^®^ solution was prepared by mixing 17.65 mL of the product with water to a final volume of 4 L. Treatments were applied using a 1.5-L backpack sprayer (Uline H-7986 Backpack Pressure Sprayer; Pleasant Prairie, WI, USA), and individual plants were sprayed until runoff, or approximately 400 mL per plant [20]. Spraying was conducted by utilizing a modified protocol of three 1.48 μM silver concentration spray applications with a 5-day interval between each application once flowers were initiated [20]. The first treatment of STS and ALESCO^®^ was conducted on 3 December 2021, when flower initials had formed. The remaining two applications occurred on 7 and 12 December 2021 [20]. Male flowers were counted after plants had reached floral maturity but before the dehiscing stage of flower development on 7 January 2022 (Figure 4). The effects of the two ethylene inhibitor treatments were compared by a 2-way ANOVA.

### 4.3. Pollen Quality Staining Assessment

Two different hemp accessions were chosen to conduct pollen quality testing (*4x* H2020-376-003, *2x* H2020-425-015), comparing the two ethylene inhibitors and their potential impacts on pollen non-abortion rate (Figure 2 and Figure 3). Fresh pollen was collected from mature, dehiscing male flowers and deposited on petri dishes. Each petri dish contained approximately 20–25 flowers. Pollen was then immediately stored at 4 °C in a sealed container containing a bed of silica beads (L2K Commerce, Brea, CA, USA) to reduce moisture within the container and stain tests in 3 days. Pollen was removed from storage and distributed on the slide for observation. Two pollen staining methods were chosen to conduct pollen quality testing. Fluorescein diacetate (FDA) (Life Technologies Corporation, Eugene, OR, USA) was utilized at a concentration of 2 mg mL^−1^ of acetone stock solution [24]. This mixture was then combined with a 0.5 M sucrose solution for staining (Figure 2) [24]. Pollen cells were counted at 10× magnification as high quality if cells plasmalemma fluoresces green [24]. Pollen cells were stained and photographed within 15 min of the application of the FDA to avoid any breakdown in inflorescence [23]. Pictures were taken using Zeiss Microscope Axio Imager.A2 (Carl Zeiss Microscopy GmbH, Jenna, Germany) with imaging software (ZEN 2 (blue edition) V1.0 en), with hand counting taking place after photo observations were taken and counted [23]. The aborted and non-aborted pollen numbers were then recorded.

A 2% acetocarmine solution (VMR Boreal Science, St. Catherines, ON, Canada) was utilized using a protocol of 40 μL per slide of pollen and incubated in a humidity chamber for 5 h [32]. Pollen was analyzed using the Zeiss Microscope Axio Imager.A2 (Carl Zeiss Microscopy GmbH, Jenna, Germany) with imaging software (ZEN 2 (blue edition) V1.0 en). Total and stained pollen were counted and analyzed to obtain pollen quality percentages [23]. Pollen non-abortion of each treatment on each genome is used to indicate the pollen quality. For each combination of ethylene inhibitor treatment, genotype, and stain method, 697 to 1371 pollen were investigated. The logistic regression general linear model (GLM) was then used to analyze the effect of ethylene inhibitor treatment and genotype on pollen non-abortion proportion.
Pollen non-abortion rate = stained pollen number/counted pollen number × 100%.

## 5. Conclusions

Traditional STS has been commonly used to stimulate male flowers from female hemp plants because of its effectiveness; however, the instability and short self-life of the chemical solutions cause extra labor and potentially inconsistent results. This research demonstrates an alternative method that uses ALESCO^®^, a commercial ethylene inhibitor-inhibit product in the flora industry, to stimulate hemp flowers. Our result indicated that ALESCO^®^ showed equal effectiveness in stimulating male flower count with improved pollen quality (non-abortion rate) compared to the traditional STS foliar spray method. Thus, ALESCO^®^ should be considered a viable alternative to the traditional STS in future breeding and feminized seed production of *C. sativa*. The observation on the difference in pollen quality between traditional STS and ALESCO^®^ might be caused by the inactive ingredients of ALESCO^®^, and they could be surfactants, wetting agents, and adjuvants that enhance silver thiosulfate to attach and penetrate plant tissues. However, the inactive ingredients of ALESCO^®^ are not labeled. Further research on adding such enhancement ingredients to the foliar spay buffer will be needed to improve the hemp male flower stimulation process and feminized seed production.

## Figures and Tables

**Figure 1 plants-12-03371-f001:**
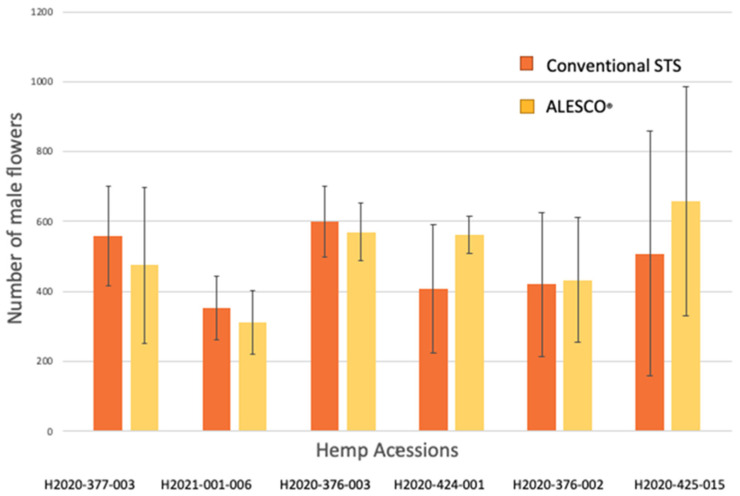
Mean and standard deviation exhibited in male flower count across six different genotypes, with three repeats per genotype in each treatment (in total 36 plants). No significant difference in flower count was discovered between the two treatments (*p* > 0.05) and between the six genotypes (*p* > 0.1) in the ANOVA analysis.

**Figure 2 plants-12-03371-f002:**
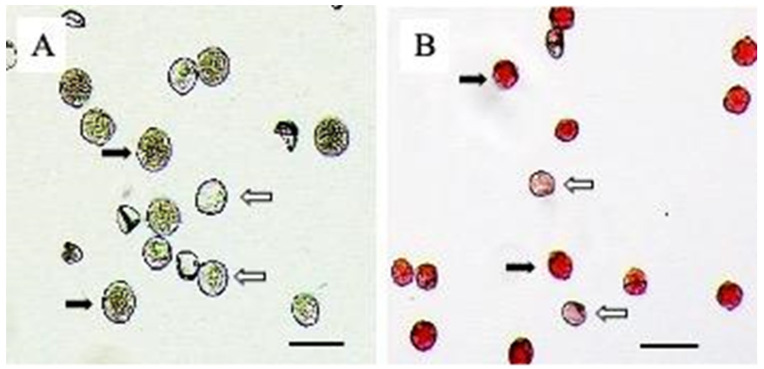
Hemp pollen grains (H2020-425-015) were stained by (**A**) fluorescein diacetate, FDA, and (**B**) 2% acetocarmine with stained pollen (non-abortion, solid arrows) and non-stained (abortion, empty arrows). Image collected using Zeiss Microscope Axio Imager.A2 with Image J ZEN 2 (blue edition) software V1.0 en. Bar = 50 μm.

**Figure 3 plants-12-03371-f003:**
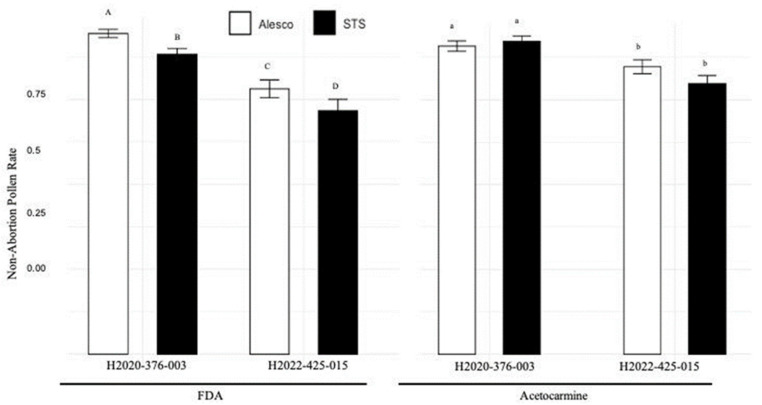
Mean and 95% confidence interval of pollen non-abortion rates. Pollen of ALESCO^®^ and STS-treated H2020-376-003 and H2022-425-015 plants. ALESCO^®^ and STS treatments. Non-abortion rates were investigated by 2% acetocarmine and fluorescein diacetate (FDA) staining. For each combination of treatment and genotype, 697–1397 pollen grains were investigated, and non-abortion rates were compared by logistic GLM analysis separately for each stain method. The alphabet with upper and lower cases indicates the two independent Tukey’s honestly significant difference tests with 95% confidence.

**Figure 4 plants-12-03371-f004:**
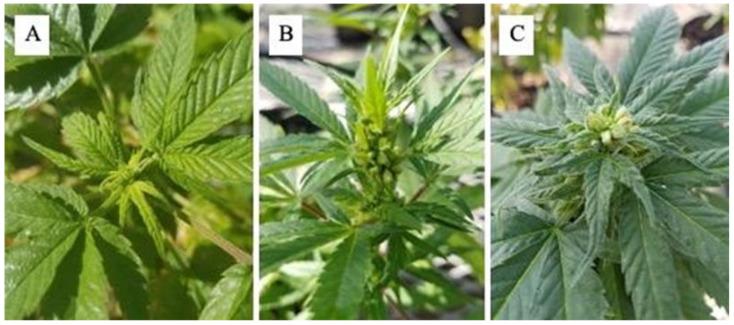
Male hemp flower cycle from initiation: (**A**) flower formation, (**B**) and dehiscing, and (**C**) pollen collection begins. On average, this process takes approximately 4–5 weeks after the last ethylene inhibitor foliar spray.

**Table 1 plants-12-03371-t001:** Plant materials, accession number, ploidies, and pedigree information.

Accession ^z^	Ploidy *	Female Parent	Male Parent
H2020-376-002	*4×*	Seedling from a bulk population converted from BaOx, Cherry Mon, T1 Iso	
H2020-376-003 ^x^	*4×*		
H2020-377-003	*4×*		
H2021-001-006	*4×*	BaOx	OP
H2020-424-001	*4×*	Seedling from a bulked population converted from T1 × OP and Cherry Mom × OP	
H2020-425-015 ^x^	*2×*	Seedling from a bulked population T1 × OP and Cherry Mom × OP	

* Ploidy tested by flow cytometry. ^z^ Used for recording the ethylene inhibitor treatment effect on male flower count. ^x^ Used for recording the ethylene inhibitor treatment effect on pollen quality.

## Data Availability

Not applicable.

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
