# Peer review of "A Novel Method for Stimulating Cannabis sativa L. Male Flowers from Female Plants"

_plants, 2023, doi:10.3390/plants12193371_

Round 1

Reviewer 1 Report

The manuscript (plants-2570387), study female hemp plants are preferred for their high cannabinoid content. To produce feminised seeds, male flowers need to be induced on these female plants. Typically, a mixture called STS (silver thiosulfate + sodium thiosulphate) is used, but it has a limited shelf life. This study explored Chrysal ALESCO®, a commercial product containing silver nitrate, for the same purpose due to its extended shelf life. In tests on hemp plants, both STS and Chrysal ALESCO® produced similar numbers of male flowers. However, Chrysal ALESCO® demonstrated superior pollen quality when tested using the FDA staining method. Overall, given its shelf-life stability and comparable results, Chrysal ALESCO® might be a better alternative for inducing male flowers on female hemp plants.

The sections of introduction, results, materials and methods, and discussion are good and informative. However, they need grammatical improvements. Some sentences are confusing and repetitive. They should be more to the point. The first issue the authors need to address is the manuscript's standardisation. The authors' instructions should be thoroughly checked. Scientific names should be italicised. Why is there such a significant variation in the values $3.00-10.00? If an abbreviation is mentioned in the text, it should be written out in full. What are the authors' objectives with this paper? This should be written at the end of the introduction. Is this a subtopic of the results? Why is it bolded? "Ethylene inhibitors impact on flower count:" Check this against the other subsections of the manuscript. Supplementary files should be in a separate file. For Figure 3's legend, what was the statistical test and how many samples were analysed? The discussion needs more robust comparisons with other data, as well as with other plants that have similarities in the development of female reproductive structures. Is this mechanism preserved in nature? For instance, how can this mechanism improve production efficiency? Standardise the naming convention with ALESCO® in superscript. "25 μmol" is incorrect. The unit should be µmol m-2 s-1. The materials and methods section should describe the cultivation conditions, analyses, and statistical tests used in detail. P values are defined but not reported in this section. Why use this fertilisation ratio? 0 N–8.7P–16.6K. I do not understand table 1. Why subdivide into Female and Male when there's no clear division, except for BaOx and OP? The conclusions need to be expanded upon. What are the future prospects? What are the next steps in cultivation and in improving this plant's productivity? How can the compounds be better used by the industry with the advancements of this study? Review references that are not recent."

Remember that this translation provides a general idea. The exact nuances might require context. Always ensure the text matches your intended meaning before finalising.

Needs correction in English, for example, in some phrases, grammar, and verbosity.

Author Response

Dear Reviwer:

We really appreciate your effort to help us improve this manuscript. We have worked on some editing thorough on your suggestions. Some specific replies were added to reply to your comment below. 

The manuscript (plants-2570387), studies female hemp plants that are preferred for their high cannabinoid content. To produce feminised seeds, male flowers need to be induced on these female plants. Typically, a mixture called STS (silver thiosulfate + sodium thiosulphate) is used, but it has a limited shelf life. This study explored Chrysal ALESCO®, a commercial product containing silver nitrate, for the same purpose due to its extended shelf life. In tests on hemp plants, both STS and Chrysal ALESCO® produced similar numbers of male flowers. However, Chrysal ALESCO® demonstrated superior pollen quality when tested using the FDA staining method. Overall, given its shelf-life stability and comparable results, Chrysal ALESCO® might be a better alternative for inducing male flowers on female hemp plants.

Thank you 

The sections of introduction, results, materials and methods, and discussion are good and informative. However, they need grammatical improvements. Some sentences are confusing and repetitive. They should be more to the point. The first issue the authors need to address is the manuscript's standardisation. The authors' instructions should be thoroughly checked. 

We have edited some sentences. Thanks 

Scientific names should be italicised. 

We have fixed this issue. Thank you for pointing it out. 

Why is there such a significant variation in the values $3.00-10.00?

We apologize for the typo. From the reference, it should be $5.00-10.00. Line 48
8. Darby, H. (2020). Industrial hemp for flower Production: A guide to basic production techniques. University of Vermont Extension Northwest Crops and Soils Program.

If an abbreviation is mentioned in the text, it should be written out in full. We have fixed this issue.

Thank you for pointing it out. Line 77-79

What are the authors' objectives with this paper? 

We have modified the sentences from lines 114-119 to address the objectives of this paper.

 Why is it bolded? "Ethylene inhibitors impact on flower count:" Check this against the other subsections of the manuscript. 

Yes, we use the bold as the sub-topics. The format is consistent for the other parts.

EX

Ethylene inhibitors impact on flower count:
Pollen quality (abortion rate) by FDA and acetocarmine:

Supplementary files should be in a separate file. 

This is fixed in the new submission. Thanks 

For Figure 3's legend, what was the statistical test and how many samples were analysed? 

This is fixed. Line 171-173

The discussion needs more robust comparisons with other data, as well as with other plants that have similarities in the development of female reproductive structures. Is this mechanism preserved in nature? For instance, how can this mechanism improve production efficiency? 

This is a great suggestion but we lack the knowledge in using ALESCO® and STS in other species for plant sex conversion purposes. The purpose of the paper is simply to provide an alternative method for the transitional STS, thus we did not go too far in the discussion part. However, we agree that we should provide further discussion on the the possible ALESCO®’s inactive ingredients that could cause the differences and that can be a direction of future research in improving the STS or ALESCO® spray. Line 233-247, 346-360

Standardise the naming convention with ALESCO® in superscript.

Thanks. This issue is fixed. Line 169, 176 

 "25 μmol" is incorrect. The unit should be µmol m-2 s-1. 

Thanks. This issue is fixed. Line 253

The materials and methods section should describe the cultivation conditions, analyses, and statistical tests used in detail. P values are defined but not reported in this section. 

More description was added. Line 200-203, 372-374, 479-480. Line 

Why use this fertilisation ratio? 0 N–8.7P–16.6K. 

The fertilization was suggested by Dr. Brian Whinker, who was the expert of greenhouse management in NCSU. 

I do not understand table 1. Why subdivide into Female and Male when there's no clear division, except for BaOx and OP? 

Plant materials were given by Dr Tom Ranney at the Mountain Horticultural Crops Research and Extension Center (MHCREC) in Mills River, NC. They have selected clones from a mix of open-pollination seeds. The original purpose of the pollination was to create a highly diverse population for a mass selection breeding strategy for the NC field environment. Thus the information is a little vague. We add a few more descriptions in the MM part. However, because they have selected clones from bulk open-pollinated populations, limited information is available. 

The conclusions need to be expanded upon. 

What are the future prospects? What are the next steps in cultivation and in improving this plant's productivity? How can the compounds be better used by the industry with the advancements of this study? 

We made some modifications and added some words about the limitations and future direction of the research.

Review references that are not recent.

We have lots of citations that are from 2019-2022. However, we understand that we could missed some newest publications. Would you be able to suggest some citations that we might be able to add?

Remember that this translation provides a general idea. The exact nuances might require context. Always ensure the text matches your intended meaning before finalising.

This manuscript was originally written by an English native speaker, and also edited by the other two native speakers. However, we did try to refine the language before the second submission. Thanks for your suggestion. 

Reviewer 2 Report

The article "A Novel Method for Stimulating Cannabis sativa L. male flowers from female plants" addresses an interesting and current subject.

The summary should summarize the results a little more and talk about the objectives of the work. The introduction provides sufficient information and includes all relevant references. The results are clearly presented except figure 2 which has poor quality. The methods are adequately described and the conclusions are supported by the results. Therefore, I am in favor of its publication with improvements in the abstract and in figure 2.

Author Response

Dear Reviwer:

We really appreciate your effort to help us improve this manuscript. We have worked on some editing thorough on your suggestions. Some specific replies were added to reply to your comment below. 

The article "A Novel Method for Stimulating Cannabis sativa L. male flowers from female plants" addresses an interesting and current subject.

The summary should summarize the results a little more and talk about the objectives of the work. The introduction provides sufficient information and includes all relevant references. The results are clearly presented except Figure 2 which has poor quality. The methods are adequately described and the conclusions are supported by the results. Therefore, I am in favor of its publication with improvements in the abstract and in Figure 2.

We add a few words to further address the purpose of the research in the abstract. 

The quality of Figure 2 has been improved. 

We also made some more edits throughout the manuscript

Thank you so much for your efforts 

Reviewer 3 Report

This is very interesting work especially for the farmers in North Carolina.   I have few comments/suggestions;

can you write the objective statement at the end of introduction?  It is there but not clear…

the conclusion is nor well written, this is not summary, plus what are the limitations and future direction….

Author Response

Dear Reviwer:

We really appreciate your effort to help us improve this manuscript. We have worked on some editing

 This is very interesting work especially for the farmers in North Carolina.   I have few comments/suggestions;

can you write the objective statement at the end of introduction?  It is there but not clear…

The objective statement has been added. Line 142-150. 

the conclusion is nor well written, this is not summary, plus what are the limitations and future direction….

We made some modifications and added some words about the limitations and future direction of the research. 

We also made some more edits throughout the manuscript

Thank you so much for your efforts 

Round 2

Reviewer 1 Report

Accept in present form

Minor changes in grammar and spelling